# Transcriptome and Metabolome Analyses Provide Insights into the Flavonoid Accumulation in Peels of *Citrus reticulata* ‘Chachi’

**DOI:** 10.3390/molecules27196476

**Published:** 2022-10-01

**Authors:** Jianmu Su, Tianhua Peng, Mei Bai, Haiyi Bai, Huisi Li, Huimin Pan, Hanjun He, Huan Liu, Hong Wu

**Affiliations:** 1Guangdong Laboratory for Lingnan Modern Agriculture, College of Life Sciences, South China Agricultural University, Guangzhou 510642, China; 2State Key Laboratory of Agricultural Genomics, BGI-Shenzhen, Shenzhen 518083, China

**Keywords:** *Citrus reticulata* ‘chachi’, transcriptome, metabolome, flavonoid

## Abstract

The quality of Chinese medicinal materials depends on the content of bioactive components, which are affected by the environmental factors of different planting regions. In this research, integrated analysis of the transcriptome and metabolome of *C. reticulata* ‘Chachi’ was performed in two regions, and three orchards were included in the analysis. In total, only 192 compounds were found in fresh peels, and among 18 differentially accumulated flavonoid metabolites, 15 flavonoids were enriched in peels from the Xinhui planting region. In total, 1228 genes were up-regulated in peels from Xinhui, including the *CHS* and *GST* genes, which are involved in the salt stress response. Overall, based on the correlation analysis of flavonoid content and gene expression in peels of *C. reticulata* ‘Chachi’, we concluded that the authenticity of the GCRP from Xinhui may be closely related to the higher content of naringin and narirutin, and the increase in the content of these may be due to the highly saline environment of the Xinhui region.

## 1. Introduction

Citri Reticulatae Pericarpium (CRP), the dried ripe pericarp of *Citrus reticulata* Blanco or its cultivars, is not only consumed as a dietary supplement in China and in other eastern countries, but is also among the most popular traditional medicinal herbs used in clinical practice for the treatment of indigestion and inflammatory syndromes of the respiratory tract [1,2]. As high-quality CRP raw materials, the pericarp of *C. reticulata* ‘chachi’ (PCRC) has been authenticated as a genuine regional drug [3,4]. *C. reticulata* ‘chachi’ has been cultivated in the Xinhui county of Guangdong province in China for hundreds of years. PCRC from Xinhui county has always been well regarded as the best national product in terms of geo-herbalism [5,6]. As a traditional Chinese herbal medicine, PCRC possesses various therapeutic activities, including strengthening the spleen and eliminating dampness and phlegm [7,8]. Recently, Liang et al. reported a possible effect of the abundant flavonoids in PCRC on the prevention and treatment of COVID-19 [9]. At present, annual market sales of PCRC from Xinhui have reached USD 145 million. Due to high market demand, *C. reticulata* ‘chachi’ has been widely introduced in Guangxi province with similar climate conditions to Guangdong. However, there are significant differences in the CRC from different regions in terms of the content of bioactive components and the pharmaceutical effect of the cultivars [10,11].

The synthesis and proper accumulation of secondary metabolites are strictly controlled in a spatial and temporal manner and are influenced by the changes in the abiotic and biotic environment [12,13]. Thus, environmental factors are crucial determinants of biosynthesis and fluctuations in plant secondary metabolites [12]. By comparing and analyzing the essential oil components of *Cinnamomum cassia* from different cultivation sites, Li et al. found that the composition of the essential oils from the *C. cassia* barks from the Guangdong and Guangxi provinces had marked differences [14].

Modern phytochemical studies have shown that the chemical constituents of PCRC are flavonoids, volatile oils, polysaccharides and alkaloids [15,16,17]. The flavonoid and bioactivity features of PCRC during storage showed that the flavonoid content changes dramatically [9,18]. The storage temperature and humidity also have an effect on the bioactive flavonoids and antioxidant activity in PCRC [19,20]. In recent years, some research has focused on the contents of bioactive components in cultivars from different planting areas. However, the original plants of used in those studies were also different [11,21]. Differential components may cause by cultivars and regions. However, it is unknown whether or how the flavonoids as the main bioactive components of PCRC are affected in different geographical areas when planting the same cultivar.

To compare the differences in the biosynthesis and accumulation of flavonoid content in PCRCs from the Guangdong and Guangxi planting regions, transcriptome and metabolome techniques were employed to identify genes related to their metabolic characteristics. Our study provided valuable data for the genes involved in the biosynthesis and accumulation of flavonoid content as well as quality evaluations and to maximize resource utilization in PCRC.

## 2. Results

### 2.1. Qualitative and Quantitative Analysis of Flavonoid Metabolites

The total flavonoid contents of the peels of *C. reticulata* ‘Chachi’ grown in three orchards were determined by spectrophotometry. The results showed that the total flavonoid contents are the difference distinguishing the three orchards. The samples from Maichong had the highest content, followed by the samples from Wubao and Daxin (Figure A1). Based on these results, the flavonoid metabolites of the citrus peels from three planting areas were investigated based on UPLC–ESI–MS/MS and databases. In the present study, 254 flavonoid metabolites were identified (Appendix A), including 110 flavonoids, 62 flavonols, 32 flavonoid C-glycosides, 21 dihydro-flavone, 10 isoflavones, 7 dihydro-flavonols, 5 tannins, 3 chalcones, 2 Proanthocyanidins, 1 flavanols, and 1 dihydro-isoflavone. Principle component analysis (PCA) uses several principal components to reveal the overall metabolic differences among the groups and the variability between intra-group samples. Two principal components, PC1 and PC2, were extracted and were 32.24% and 21.08%, respectively (Figure A2). In the PCA score plot, the Maichong planting region (MC), Wubao planting region (WB), and Daxin planting region (DX) were clearly separated, and the repeated samples were gathered together compactly.

Differential flavonoid metabolites were screened for each comparison group by combining the fold changes and variable importance in project (VIP) values of the OPLS-DA model. There were 48 significantly different flavonoid metabolites between MC and DX (29 down-regulated and 19 up-regulated) (Figure 1a), and 37 between WB and DX (28 down-regulated and 9 up-regulated) (Figure 1b). After taking an intersection of each comparison group in a Venn diagram (Figure A3), 18 common differential flavonoid metabolites were observed, 15 flavonoid metabolites, including isohemiphloin, naringin, and narirutin, were up-regulated in the MC and WB of the Xinhui genuine region, and 3 flavonoid metabolites were up-regulated in the DX of other regions (Figure 1c).

### 2.2. Differentially Expressed Genes between the Three Planting Regions

To investigate the molecular processes and genes involved in flavonoid biosynthesis in peels of *C. reticulata* ‘chachi’, we compared the RNA-seq data derived from citrus peels from the three planting regions. Genes with adjusted *p*-values ≤ 0.05 and absolute log2fold change ≥ 1 were considered as significantly differential expression genes (DEGs) and are shown in volcano plots (Figure 2a,b and Figure A4). Comparative transcriptome analysis revealed that the Maichong (MC) vs. Daxin (DX) comparison generated the greatest number of DEGs (2971), followed by the Wubao (WB) vs. Daxin (DX) comparison (2504), and the Maichong (MC) vs. Wubao (WB) comparison (95). More specifically, a total of 2971 genes were differentially expressed in the MC vs. DX comparison, including 1199 up-regulated genes and 1772 down-regulated genes. A total of 2504 DEGs were identified in the WB vs. DX comparison: 991 DEGs were up-regulated and 1513 DEGs were down-regulated. In the WB vs. MC comparison, there were only 96 DEGs, 51 of which were up-regulated and 45 were down-regulated (Figure A4).

In order to find genes up-regulated in peels from the Xinhui genuine region, a intersect gene set was performed, and then the genes were used to perform pathway enrichment. The result showed that 1226 genes were co-up-regulated in MC and WB. The functional groups with a *p* value ≤ 0.05 are shown (Figure 2c), where the most enriched genes are those related to photosynthesis, photosynthesis-antenna protein, glycerolipid metabolism, carbon metabolism, carbon fixation in photosynthetic organisms, and circadian rhythm in plants. Among the 1226 differential genes, 77 genes were annotated to 28 transcription factors (TFs), which were clustered into two groups based on the Euclidean distance of gene expression (Figure A5). In the green group, we found that the expression of NAC was the highest, with an average FPKM of 377.8. This was followed by MYB, TUB, C2C2-CO-like, bHLH and Alfin-like, the FPKM values of which were 163.68, 110.73, 106.62, 75.84, and 68.65, respectively. The purple group was further divided into two subgroups, one group containing two TFs (C2CE-GATA, SBP), and the other group containing 19 TFs, such as GRAS, mTEFR, and HSF.

### 2.3. WGCNA Identifies Candidate Genes Associated with Flavonoid content

To further investigate the relationship between gene expression and metabolism, a WGCNA was performed. After clustering the samples, all of the samples were retained in the subsequent analysis (Figure 3a). Fifteen flavonoid content that were enriched in MC and WB are shown in the heatmap, and the sample clustering result showing gene expression exhibited a similar pattern with that of the flavonoid content. A soft threshold power of 9 (β = 9) was selected according to the preconditions of approximate scale-free topology (Figure A6). The analysis identified 31 distinct co-expression modules (labeled with diverse colors) which are shown in the dendrogram (Figure A7), with brown module (containing 1445 genes) being found to be positively associated with 12 flavonoids (Figure 3b), examples of which include genistein-8-C-glucoside (R = 0.87, *p* = 0.002), apigenin-5-O-glucoside (R = 0.86, *p* = 0.003), isohemiphloin (R = 0.91, *p* = 1 × 10^−5^, luteolin-4′-O-glucoside (R = 0.82, *p* = 0.007), luteolin-3′-O-glucoside (R = 0.83, *p* = 0.005), lpigenin-8-C-(2″-xylosyl) glucoside (R = 0.88, *p* = 0.002), genistein-8-C-apiosyl (1→6)glucoside (R = 0.89, *p* = 0.001), apigenin-6-C-glucoside-8-O-xylcoside (R = 0.86, *p* = 0.003), apigenin-6-C-(2″-glucuronyl) xyloside (R = 0.85, *p* = 0.004), vitexin-2″-O-rhamnoside (R=0.87, *p* = 0.003), and naringenin-7-O-rutinoside (narirutin) (R = 0.86, *p* = 0.003).

Considering that the brown module had a strong positive correlation with most flavonoids, we performed pathway enrichment analysis based on the genes in the module. In general, the genes in the brown module were enriched in glycolysis/Gluconeogenesis, mRNA surveillance pathway, ribosome, pyruvate metabolism, pantothenate and CoA biosynthesis (Figure 4a). Additionally, a total of 133 genes were discovered and were subsequently defined as hub genes, with a principles of adjacency threshold > 0.4. A co-expression network was constructed based on the adjacency between the hub genes in the brown module. We found that 24 genes intersected with the DEGs described above (Figure 4b). In the network, a dihydrolipoamide acetyltransferase (*DLAT*) and an unknown function gene Cs3g11540 showed a high correlation with other genes. The *DLAT* gene was found to be up-regulated in DX. Transcript factors such as MYB, C2C2-Dof, and Alfin-like were also found in the gene network. In the genes, two genes *CYP73A* and *CHS* encoding the enzymes of the phenylpropanoid biosynthesis pathway and the flavonoid biosynthesis pathway, respectively, were up-regulated in MC and WB (Figure 4c), implying that these two up-regulated genes may likely lead to higher of naringin accumulation.

The blue module (containing 3156 genes) was positively associated with 2,4,2′,4′-tetrahydroxy-3′-prenylchalcone (R = 0.88, *p* = 0.002), astilbin (R = 0.77, *p* = 0.02), naringenin-7-O-neohesperidoside (naringin) (R = 0.84, *p* = 0.004) and naringenin-7-O-rutinoside (narirutin) (R = 0.79, *p* = 0.01). Considering that naringin and its aglycone naringenin belong to this series of flavonoids and display strong anti-inflammatory and antioxidant activities. We focused on the correlations of naringin with gene modules and identified hub genes that were strongly correlated with naringin. Genes expressed in 80% of the samples and FPKM values > 1 were retained for WGCNA unsigned co-expression network analysis. In total, 108 hub genes were obtained by screening genes correlated with naringin (|GTS| > 0.9 and |GMM| > 0.9) in the blue module (Figure 5a). There were 23 hub genes that were enriched in DX and 85 hub genes that were enriched in MC and WB (Figure 5b). A co-expression network was constructed based on the adjacency > 0.4 between hub genes in the blue module. Twelve genes showed high correlation with other genes (Figure 5c), in which three genes, Cs8g20220 (protein glucosyltransferase), Cs2g14860 (unknown function) and Cs6g01210 (K01648-ATP citrate (pro-S)-lyase), were enriched in Daxin, and nine genes, Cs7g27960 (glutathione S-transferase), Cs6g02730 (ribD, diaminohydroxy-phosphoribosylamino-pyrimidine deaminase), Cs8g10800 (unknown function), Cs8g01700 (unknown function), Cs7g31190 (light-harvesting complex II chlorophyll a/b binding protein 4), orange1.1t02492 (unknown function), Cs7g31690 (unknown function) and Cs7g03460 (ribulose-bisphosphate carboxylase/fructose-bisphosphate aldolase-lysine N-methyltransferase), were enriched in MC and WB (Figure 5b). Two MYB genes, three bHLH genes and a GRAS gene showed a high correlation with these twelve genes. Interestingly, among the 12 hub genes, we found a gene that encodes glutathione S-transferase, which is widely found in citrus. In total, there were 40 glutathione S-transferase genes that were found in our study (Figure A8), in which 23 genes were up-regulated in MC and WB, and 17 genes were up-regulated in DX.

In order to explore the abiotic stress factors that cause the differences in the flavonoid content in peels of *C. reticulata* ‘Chachi’ from different regions, the salinity content of the rhizosphere soil in which *C. reticulata* ‘Chachi’ is cultivated in the three regions was measured. The results show that the soil salinity of WB was higher than that of DX, and the differences between WB and DX were significant (Figure A9, *t* test *p* = 0.050). However, the differences between the soil salinity of WB and MC were not significant (*t* test *p* = 0.379). The soil salinity of MC was a little higher than that of DX, but the differences were not significant. 

### 2.4. Salt Stress Plays an Important Role in the Accumulation of Flavonoids

To further verify whether salt stress may promote the biosynthesis of flavonoids in plants, a tobacco salt stress treatment was performed (Figure 6a). As a result, the fresh weight of the aerial part of tobacco under salt stress was significantly higher than that of the control (Figure 6b, *t* test *p* = 7.12 × 10^−5^), while the height of tobacco plant was not significant between control and treatment (Figure 6c, *t* test *p* = 0.381). Especially, the flavonoid contents in leaves of tobacco under salt stress were higher than in control (Figure 6d, *t* test *p* = 0.042). These results further verified our hypothesis that salt stress may increase the content of total flavonoids in plants.

### 2.5. Verification of Gene Expression Patterns by qRT-PCR

In order to verify the accuracy of the RNA-seq results, we selected six genes that were involved in biosynthesis of naringin for qRT-PCR analysis. The results showed that the expression pattern of these six differential genes was similar to that of RNA-seq (Figure 7). The results indicated that the RNA-seq was reliable.

## 3. Discussion

PCRC produce flavonoids as secondary metabolites, which are extensively distributed and play vital roles in plant physiology, as well as exhibit high bioactivity for clinical applications [22,23]. In the present study, the differences in the transcripts and metabolites in the samples from three different orchards, MC, WB and DX were compared. UPLC–MS/MS and untargeted metabolomics analysis were employed to intensively investigate the composition of the flavonoids in PCRC grown in different planting regions. Up to 254 metabolites of flavonoids were detected in all samples, including 110 flavonoids, 62 flavonols, 32 flavonoid C-glycosides, 21 dihydroflavone, 10 isoflavones, 7 dihydroflavonol, 5 tannins, 3 chalcones, 2 proanthocyanidins, 1 flavanol, and 1 dihydroisoflavone. Based on the comparison of the metabolites from three orchards, 18 common differential flavonoid metabolites were observed, 15 flavonoid metabolites were from the Xinhui genuine region, and three flavonoid metabolites were up-regulated in DX. These results imply that peels from Xinhui contain a higher flavonoid content. 

Transcriptome analysis results revealed 1226 differentially expressed genes in MC and WB. These differentially expressed genes were assigned to the photosynthesis, photosynthesis-antenna proteins, glycolipid metabolism, and carbon metabolism pathways. Flavonoid biosynthesis is light-dependent, and higher light intensities stimulate the synthesis of phenols and flavonoids to protect living plants [24,25]. Greater exposure to sunlight increases flavonoid contents and influences the color and nutritional quality of the fruit [26]. The pathway enrichment of differentially expressed genes suggests that the accumulation of flavonoids in citrus peels may be likely related to the amount of light received by the plant.

Using WGCNA analysis, we identified 31 distinct co-expression modules, and the brown module was found to be positively associated with 12 flavonoids (Figure 3b). The genes in the brown module were assigned to the glycolysis-gluconeogenesis mRNA surveillance pathway and to ribosome (Figure 4a). *DLAT* is an enzyme component of the multienzyme pyruvate dehydrogenase complex that is responsible for the pyruvate decarboxylation step that links glycolysis to the citric acid cycle, and it is closely related to flavonoid biosynthesis [27]. Additionally, the repression of dihydrolipoamide acetyltransferase (DLAT) led to improvements in the concentration of naringenin [28], which is a substrate of many flavonoids [29]. A co-expression network showed a high correlation between DLAT and Cs3g11540 (gene with an unknown function) with other genes. Transcription factors such as MYB, C2C2-Dof, and Alfin-like were also found in the hub gene network, suggesting that these TFs are an important regulator of flavonoid biosynthesis, and play a crucial role in flavonoid synthesis in citrus [30,31].

Chalcone synthase (*CHS*) catalyzes the first and most important step of in flavonoid biosynthesis by directing carbon flux from general phenylpropanoid metabolism to the flavonoid pathway [32]. The overexpression of EaCHS1 increases the production of downstream flavonoids and the expressions of related genes in the phenylpropanoid pathway [33,34]. *CHS* was discovered to be up-regulated in the MC and WB of Xinhui, Guangdong; this is thought to increase the accumulation of flavonoids in the PCRC from Xinhui. Out of 40 glutathione S-transferase (*GST*) genes, 23 genes were up-regulated in MC and WB, and 17 genes signaling were up-regulated in DX. It has been reported that *GST* is involved in *CHS* signaling in the plant cell and plays an important role in tolerance to salt stress [35,36,37]. The up-regulation of *CHS* found in MC and WB implies that the *GST* genes were in high correlation with *CHS*. Salt stress can increase the accumulation of antioxidant flavonoids in the leaves of capsicum and cyanobacteria [38,39]. In our result, we found that the soil salinity of WB and MC was higher than that of DX. The salt stress of tobacco showed that the total flavonoids of treatment were higher than control. These results suggest that the difference in flavonoid content in the peels of *C. reticulata* ‘Chachi’ between different regions may be caused by salt stress.

In summary, the peels from Xinhui contain a higher content of flavonoids, which may be caused by salt stress induced by the soil conditions of the Xinhui region. Firstly, salt stress may repress the *DLAT* expression level, which may lead to improvements in the concentration of naringenin. Some transcription factors such as MYB, C2C2-Dof, and Alfin-like likely play an important role during this process. Secondly, *GST* plays an important role in this salt stress process. Salt stress promotes the up-regulation of *CHS*, thereby further promoting the biosynthesis of naringenin. 

## 4. Materials and Methods

### 4.1. Plant Materials

Two representative orchards (Maichong: 22°24′ N, 113°1′ E and Wubao: 22°28′ N, 112°57′ E) with prominent *C. reticulata* ‘Chachi’ production in Xinhui county (geo-authentic regional product), Guangdong province, and one orchard (Daxin: 23°17′ N, 117°27′ E) in Guangxi province were selected. For each orchard, the peels of three citrus trees were collected for transcriptome sequencing and untargeted metabolomic profiling. Peels were frozen in liquid nitrogen immediately and were stored at −80 °C until further use.

### 4.2. RNA Isolation and Transcriptome Sequencing

RNA from the peels was extracted using a Column Plant RNAout Kit according to the instructions (TIANDZ, Beijing, China). RNA extraction was quantified and assessed for integrity using NanoDrop (Thermo, Massachusetts, USA) and a 2100 Agilent Bioanalyzer (Agilent, California, USA) prior to subsequent experiments. To construct the BGI-based mRNA-seq library, 1 ug qualified RNA from each sample was used. Then, the library was sequenced on the BGISEQ platform using a 100 bp pair-ended strategy.

The raw reads were filtered by SOAPfilter [40], and an average of 4 Gb high-quality data were generated for each sample. Then, the resulting high-quality reads from each sample were mapped against the citrus sinensis reference genome [41] using hisat2 [42]. The expression level of each gene in each sample was calculated as the FPKM with StringTie2 [43]. The differential expression genes were identified using DEseq2 [44] based on the read’s matrix, with an FDR *p* value correction value < 0.05 and absolute fold change value > 2. The differential genes were enriched into pathways using Fisher’s exact test.

### 4.3. Identification of Co-Expression Modules and Hub Genes

A gene co-expression network was built using the WGCNA package (Version 4.0.2) [45] in R (Alcatel-Lucent Bell, Labs New Jersey, USA). Genes expressed in 80% of samples and with FPKM values > 1 were retained for WGCNA unsigned co-expression network analysis. Parameters were set up as power = 9, minModuleSize = 30, and MEDissThreshold = 0.25. A total of 31 different modules were derived. To construct co-expression genes and metabolites networks, genes in the modules correlated with naringin were used. Hub genes were obtained by screening genes correlate with metabolites (|GTS| > 0.9 and |GMM| > 0.9). The hub gene co-expression network and the resulting gene–gene interactions were used to visualize the subnetworks using Gephi (Version 0.92) [46].

### 4.4. Metabolic Profiling

The freeze-dried peels were crushed using a mixer mill (MM 400, Retsch, Haan, Germany) with a zirconia bead for 1.5 min at 30 Hz. Next, 100 mg powder was weighted and extracted overnight at 4 °C with 1.0 mL of 70% aqueous methanol. Following centrifugation at 10,000× g for 10 min, the extracts were absorbed and filtrated before LC–MS analysis.

The sample extracts were analyzed using an LC–ESI–MS/MS system (HPLC, Shim-pack UFLC SHIMADZU CBM30A system, www.shimadzu.com.cn/, accessed on 20 June 2021; MS, Applied Biosystems 4500 Q TRAP, www.appliedbiosystems.com.cn/, accessed on 20 June 2021). LIT and triple quadrupole (QQQ) scans were acquired on a triple quadrupole-linear ion trap mass spectrometer (Q TRAP), API 4500 Q TRAP LC/MS/MS System, equipped with an ESI Turbo Ion-Spray interface, operating in a positive ion mode.

### 4.5. Tobacco Salt Stress Treatment

Seeds of tobacco cultivars, Nicotiana tabacum, were planted into pots filled with soils. Tobacco plants were transplanted to a single pot after three weeks growing. Three days later, 0 mM/L (control, watering) and 50 mM/L NaCl (salt stress) were applied to each pot for 14 days. Each group has five independent plants as replicates. Total flavonoids were extracted via the approach descript above. The content of total flavonoids in the extract of tobacco leaves at the wavelength of 510 nm by UV and rutin as the standard.

### 4.6. Validation of the RNA Sequence Data Using qRT-PCR

The RNA was extracted as stated before. The RNA extraction was quantified and assessed for integrity using the NanoDrop (Thermo, Massachusetts, USA), and was reverse transcribed into cDNA using the PrimeScript RT reagent Kit (TaKaRa, Dalian, China). The expression levels of the six genes involved in the biosynthesis of naringin were measured by quantitative real-time PCR using SYBR Green Supermix on the CFX96 Real-Time PCR Detection System (BIO-RAD, California, USA) according to the manufacturer’s protocol. Each sample had three biological replicates with three technical replicates for each biological replicate. The relative expression level was calculated by the equation ratio 2^−ΔΔCt^. The primers of these genes were designed using primer 6 software (PREMIER Biosoft International, San Francisco, USA) and primers can be seen in Table 1.

### 4.7. Data Analysis

Unsupervised PCA (principal component analysis) was performed by the statistics function prcomp within R (www.r-project.org). The data were unit variance scaled before unsupervised PCA. Significantly regulated metabolites between groups were determined by VIP ≥ 1 and absolute log2FC (fold change) ≥ 1. VIP values were extracted from the OPLS-DA results, which also contained score plots and permutation plots, and were generated using the R package MetaboAnalystR [47]. The statistical analysis was carried out using Student’s unpaired t-test to compare the total content of flavonoids of tobacco leaves. Effects are considered significant at *p* < 0.05.

## 5. Conclusions

We performed an integrated transcriptome and metabolome analysis on *C. reticulata* ‘Chachi’ peels collected from three orchards in the Guangdong and Guangxi provinces. A regulatory network of metabolites and transcripts involved in flavonoid biosynthesis and salt stress was revealed. We identified some candidate genes involved in flavonoids accumulation, including two MYB genes and three bHLH genes that are most likely involved in the regulation of flavonoids biosynthesis, and one glutathione S-transferase gene involved in salt stress that is probably promoted for flavonoids accumulations. Taken together, we concluded that PCRC from Xinhui region has high contents of naringin and narirutin, which may be due to the high-salt environment in Xinhui region. The findings provide an avenue for further investigation into the biosynthesis and genetic regulation of citrus flavonoid metabolism and may aid in the selection of germplasms with a richflavonoid content. 

## Figures and Tables

**Figure 1 molecules-27-06476-f001:**
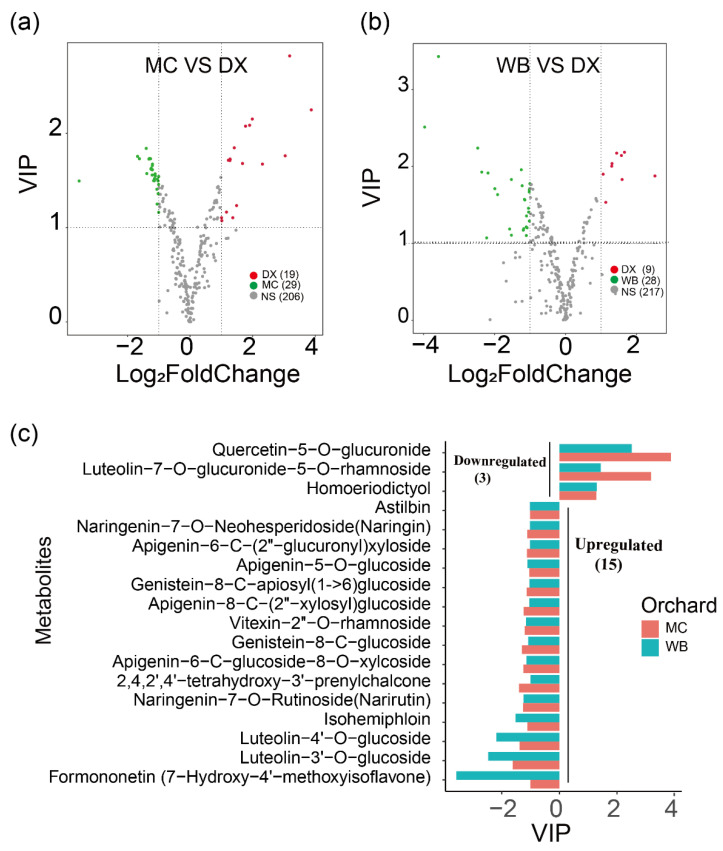
Differences in flavonoid content among three orchards. (**a**) Comparison of metabolite differences between Maichong (MC) orchard and Daxin (DX) orchard. The red dots represent the up-regulated metabolites in Daxin orchard, the green dots represent the up-regulated metabolites in Maichong orchard, the grey dots represent the metabolites with no significance, and the numbers in brackets are the metabolite numbers. VIP, variable importance in projection. (**b**) Comparison of metabolite differences between Wubao (WB) orchard and Daxin (DX) orchard. The abscissa is the log value of the fold change, and the ordinate is the VIP value. The metabolites with |log (fold change)| ≥ 1 indicate differential metabolites. The red dots represent the up-regulated metabolites in Daxin orchard, the green dots represent the up-regulated metabolites in Wubao orchard, the grey dots are the metabolites with no difference, and the numbers in brackets are the metabolite numbers. (**c**) The abscissa is VIP, and the ordinate is the metabolites. The red bar represents the up-regulated or down-regulated metabolites from Maichong, and blue represents the up-regulated or down-regulated metabolites from Wubao. VIP ≤ −1 indicates that the metabolite up-regulated by Maichong or Wubao, VIP ≥ 1 means the metabolite is down-regulated in Maichong or Wubao.

**Figure 2 molecules-27-06476-f002:**
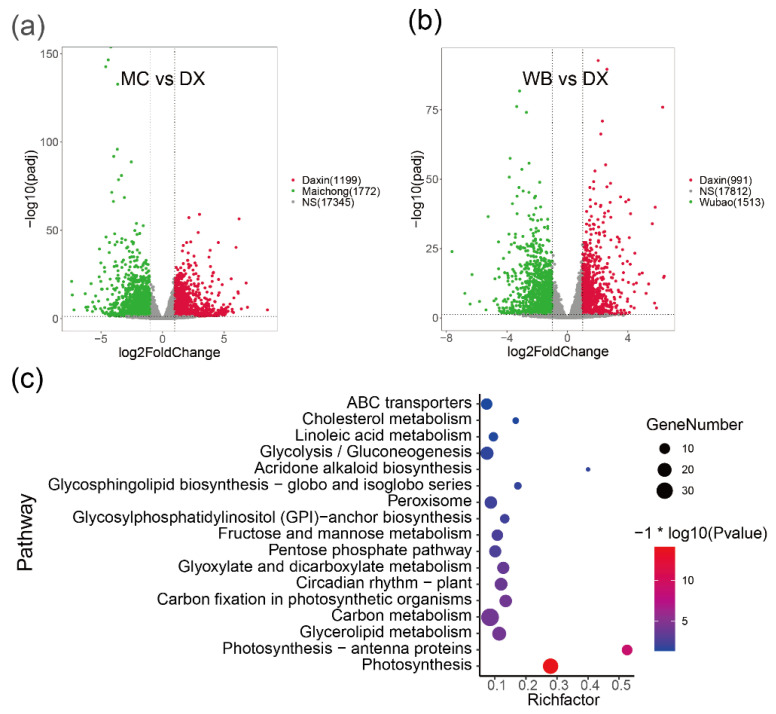
Volcano plots of DEGs in the pairwise comparisons and pathway enrichment. (**a**) Volcano plot of DEGs in the MC vs. DX comparison. Green dots represent genes up-regulated in Maichong. Red dots represent genes up-regulated in Daxin. Grey dots represent genes that show no significant difference between two orchards. (**b**) Volcano plot of DEGs in the WB vs. DX comparison. Green dots represent genes up-regulated in Wubao. Red dots represent genes up-regulated in Daxin. Grey dots represent genes that are not significantly different between two orchards. (**c**) Pathway enrichment of genes co-up-regulated in MC and WB. The pathway with *p* value ≤ 0.05 are shown. Node size represents the number of differential genes within the pathway and color represents the significance of the pathway. ‘Gene number’ is the number of genes enriched in a pathway. ‘Rich factor’ is the percentage of total DEGs in the given pathway term.

**Figure 3 molecules-27-06476-f003:**
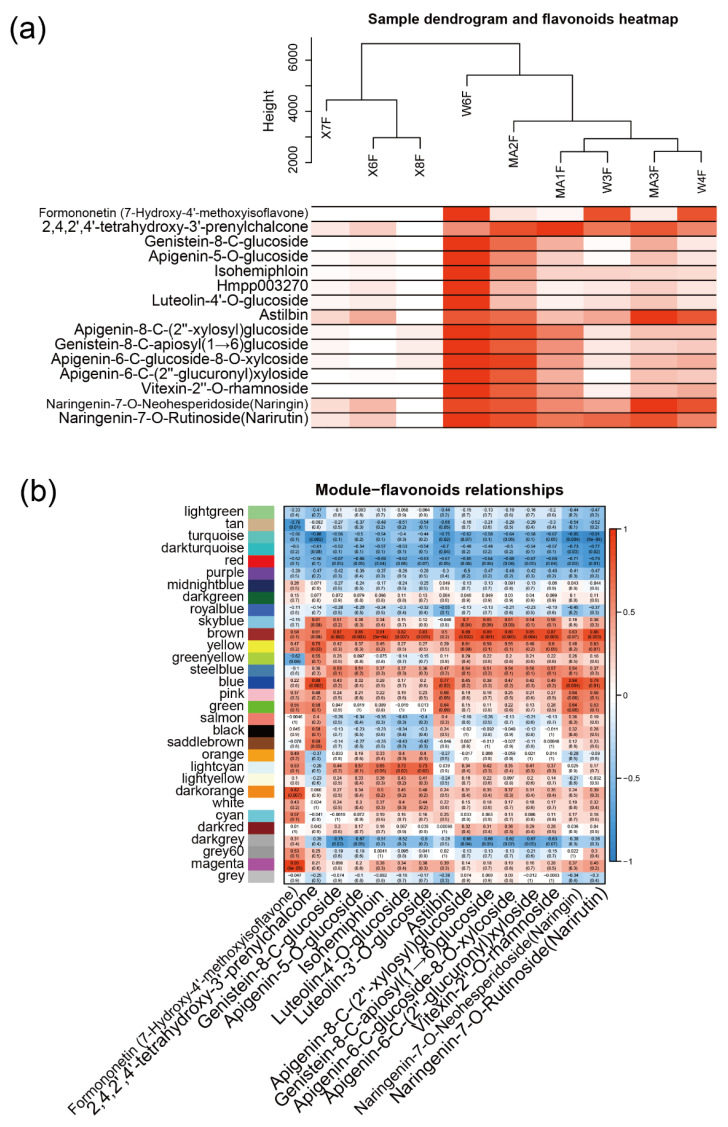
WGCNA network and module–trait correlation analysis. (**a**) Sample dendrogram and flavonoid content heatmap. Color shade of white to red represents low to high of the content of flavonoids. (**b**) Correlations of flavonoids with WGCNA modules. Each row corresponds to a module and is labeled with the same color as in Figure A7. The columns correspond to flavonoids. The color of each cell indicates the correlation coefficient between the module and the flavonoids (the top number in the cell represents the correlation coefficient, and the bottom one in parentheses represents the *p* value).

**Figure 4 molecules-27-06476-f004:**
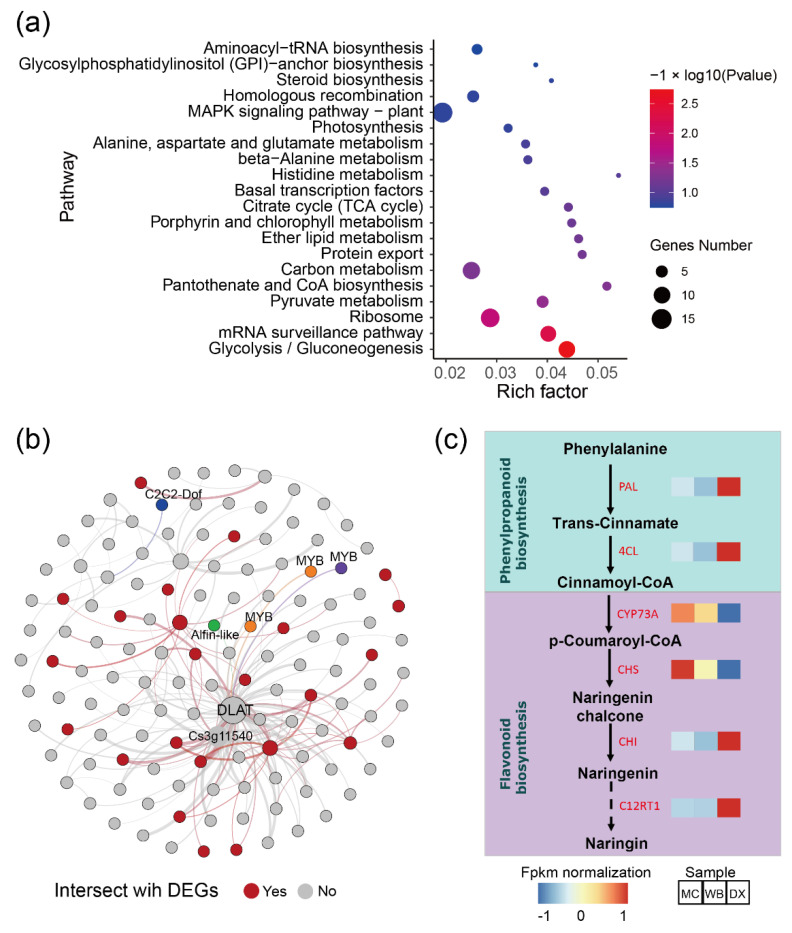
Co-occurrence network analysis based on the genes in module brown. (**a**) Pathway enrichment of genes in the brown module (*p* value ≤ 0.05 are shown). Node size represents the number of differential genes within the pathway and color represents the significance of the pathway. ‘Gene number’ is the number of genes enriched in a pathway. ‘Rich factor’ is the percentage of total DEGs in the given pathway term. (**b**) Co-occurrence network analysis based on hub genes with the principles of adjacency threshold > 0.4. Nodes that are red represent genes intersecting with DEGs between MC and WB vs. DX. Genes annotated to TFs are marked. Node size represents the degrees of the genes in the network. (**c**) Gene expression in the biosynthesis of naringin. FPKM was normalized and is shown in color. *PAL*, phenylalanine ammonia-lyase. *4CL*, 4-coumarate-CoA ligase. *CYP73A*, trans-cinnamate 4-monooxygenase. *CHS*, chalcone synthase. CHI, chalcone isomerase. *C12RT1*, flavanone 7-O-glucoside 2″-O-beta-L-rhamnosyltransferase.

**Figure 5 molecules-27-06476-f005:**
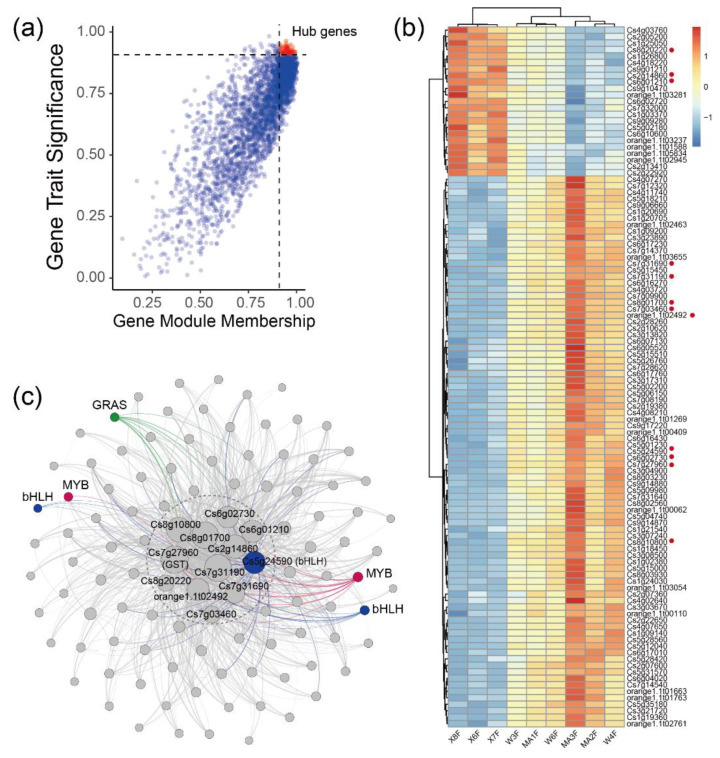
Identification of hub genes in blue modules. (**a**) The correlation of gene module membership and gene trait significance. Hub genes were obtained by screening genes which correlate with naringin (|GTS| > 0.9 and |GMM| > 0.9) in the blue module. Red dots represent hub genes. (**b**) FPKM values of 108 hub genes in the heatmap. FPKM was centered and scaled in the row. (**c**) Co-occurrence network analysis based on 108 hub genes with the principles of adjacency threshold > 0.4. Genes annotated to TFs were marked. Node size represents the degrees of the genes in the network.

**Figure 6 molecules-27-06476-f006:**
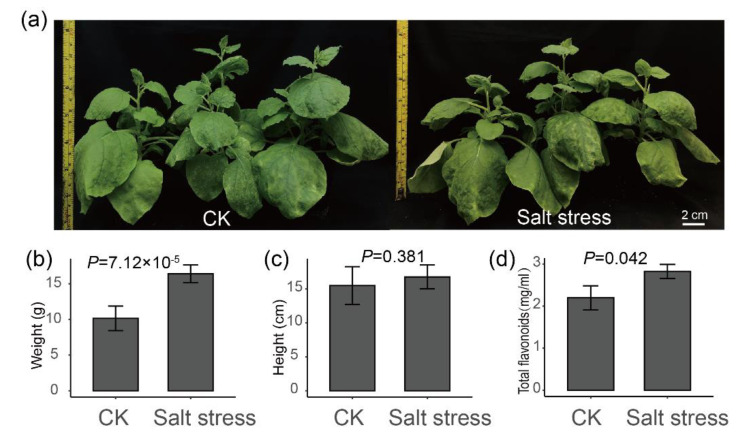
Plant height, fresh weight and total flavonoid content of tobacco under salt stress. (**a**) The phenotypic changes in tobacco plant after 14 days of salt stress. (**b**) The fresh weight of the aerial part of tobacco. (**c**) The height of the tobacco. (**d**) Total flavonoid content of tobacco.

**Figure 7 molecules-27-06476-f007:**
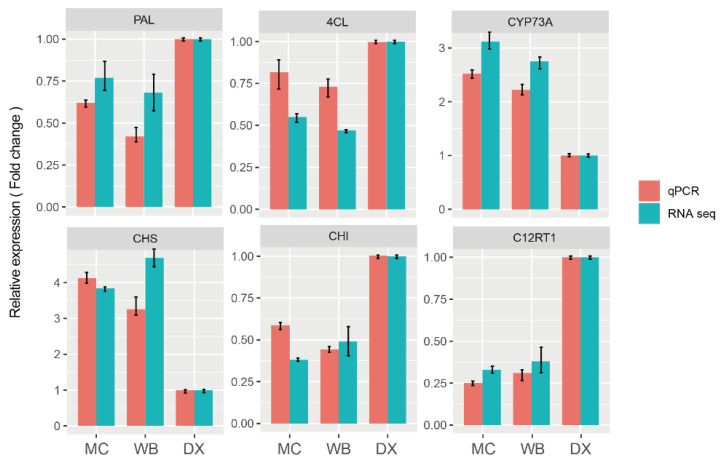
qRT-PCR validation profiles of six selected genes. The expression levels of genes in DX were used as the reference state, which was set to 1, and fold change values are shown here.

**Table 1 molecules-27-06476-t001:** The primer of six genes.

Gene	Former Primer	Reverse Primer
*PAL*	5′-ACGTTGAAGGGAGCCATTT-3′	5′-ACGTTGAAGGGAGCCATTT-3′
*4CL*	5′-TTCCGATCAAAGTTGCCCGA-3′	5′-CGGCAGCGAATTTACGTGAG-3′
*CYP73A*	5′-AGGGAGTGGAGTTCGGATCA-3′	5′-CGTGAAGAAAGGGACGGTCA-3′
*CHS*	5′-GGCCTCAACCCATCTGTCAA-3′	5′-AGCATACGACTAGAACGCGG-3′
*CHI*	5′-ATGCAAGTCCCAGTGCATCA-3′	5′-TTGGGCAACGGATTGCAAAG-3′
*C12RT1*	5′-GAGAGAGGTGGCCGACATTA-3′	5′-TCTTCGCACACTTGGATGCC-3′

## Data Availability

The data presented in this study are available within this article and its Appendix B and Appendix A.

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
