# Peer review of "Transcriptome and Metabolome Analyses Provide Insights into the Flavonoid Accumulation in Peels of Citrus reticulata ‘Chachi’"

_molecules, 2022, doi:10.3390/molecules27196476_

Round 1

Reviewer 1 Report

The manuscript “Transcriptome and metabolome analyses provide insights into the flavonoid accumulation in peels of Citrus reticulata ‘Chachi’” identify that 192 compounds were found in fresh peels, and 15 flavonoids were enriched in peels from Xinhui planting region in two regions including three orchards. The results are well presented with a lot of interesting information. However, there are some aspects of the manuscript that need to be modified and improved. The comments for the manuscript are as follows:

1. The manuscript should be improved by English polishing.

2. In Introduction, the references on Citri Reticulatae Pericarpium metabolome in recent years were not listed in the text. The purpose of the manuscript was not clearly state.

3. In 2.3. WGCNA identifies candidate genes associated with flavonoids content, WGCNA was used to investigate the relationship between gene expression and metabolism, if the data of environmental factor should be given in the text?

4. The transcriptome data should be confirmed by qPCR.

5. In 2.4. Salt stress plays an important role in the accumulation of flavonoids, line 224-230 should be placed in the paragraph 2.3 or deleted.

6. Please add scale bar in the figure 6a.

7. The style of references should be check one by one with the Journal Instruction.

Reviewer 2 Report

The article contains original data obtained by modern transcriptome and LC-ESI-MS/MS techniques. The purpose of the above research is missing; did the authors mean natural product standardization? There is no description of the statistical methods used (in the Methods section). The last point of the discussion is probably the reviewer's suggestions (lines 300-303). In my opinion, there should be an analysis of the soil from the habitats from which the plants were obtained. The description of the methodology lacks the experiment illustrated in Figure 6.

Round 2

Reviewer 1 Report

The manuscript is revised carefully, while there is also some questions should be addressed.

1. Line 48,49, C. cassia should be in italic or not?

2. Line 170: The first letter of Rutinoside(Narirutin) should be in capital letter? There should be a space between Rutinoside and (Narirutin). Please check it in the text.

3. The genes MYB, C2C2-Dof, and Alfin-like should be in italic or not? Please keep in the same style.

4. Line 342: Please keep the same style of ‘Chachi’ in the text.

5. Reference: Please check the style of references one by one.

Reviewer 2 Report

The Authors addressed each concern. In my opinion, their MS is suitable for publication. Accept.